# Nilotinib: A Tyrosine Kinase Inhibitor Mediates Resistance to Intracellular Mycobacterium Via Regulating Autophagy

**DOI:** 10.3390/cells8050506

**Published:** 2019-05-26

**Authors:** Tariq Hussain, Deming Zhao, Syed Zahid Ali Shah, Naveed Sabir, Jie Wang, Yi Liao, Yinjuan Song, Haodi Dong, Mazhar Hussain Mangi, Jiamin Ni, Lifeng Yang, Xiangmei Zhou

**Affiliations:** 1Key Laboratory of Animal Epidemiology and Zoonosis of the Ministry of Agriculture, College of Veterinary Medicine, China Agricultural University, Beijing 100193, China; drtariq@aup.edu.pk (T.H.); zhaodm@cau.edu.cn (D.Z.); zahidvet@cuvas.edu.pk (S.Z.A.S.); naveedsabir@upr.edu.pk (N.S.); wangjie1985abc@163.com (J.W.); liaoyi_cau@126.com (Y.L.); syinjuan@126.com (Y.S.); dhd0905@cau.edu.cn (H.D.); drmazharmangi114@gmail.com (M.H.M.); jiamin1014@cau.edu.cn (J.N.); yanglf@cau.edu.cn (L.Y.); 2Department of Pathology, Faculty of Veterinary Science, Cholistan University of Veterinary and Animal Sciences, Bahawalpur 63100, Pakistan

**Keywords:** nilotinib, *Mycobacterium bovis* (*M. bovis*), *Mycobacterium avium* subspecies *paratuberculosis* (MAP), macrophage, autophagy

## Abstract

Nilotinib, a tyrosine kinase inhibitor, has been studied extensively in various tumor models; however, no information exists about the pharmacological action of nilotinib in bacterial infections. *Mycobacterium bovis* (*M. bovis*) and *Mycobacterium avium* subspecies *paratuberculosis* (MAP) are the etiological agents of bovine tuberculosis and Johne’s disease, respectively. Although *M. bovis* and MAP cause distinct tissue tropism, both of them infect, reside, and replicate in mononuclear phagocytic cells of the infected host. Autophagy is an innate immune defense mechanism for the control of intracellular bacteria, regulated by diverse signaling pathways. Here we demonstrated that nilotinib significantly inhibited the intracellular survival and growth of *M. bovis* and MAP in macrophages by modulating host immune responses. We showed that nilotinib induced autophagic degradation of intracellular mycobacterium occurred via the inhibition of PI3k/Akt/mTOR axis mediated by abelson (c-ABL) tyrosine kinase. In addition, we observed that nilotinib promoted ubiquitin accumulation around *M. bovis* through activation of E3 ubiquitin ligase parkin. From in-vivo experiments, we found that nilotinib effectively controlled *M. bovis* growth and survival through enhanced parkin activity in infected mice. Altogether, our data showed that nilotinib regulates protective innate immune responses against intracellular mycobacterium, both in-vitro and in-vivo, and can be exploited as a novel therapeutic remedy for the control of *M. bovis* and MAP infections.

## 1. Introduction

*Mycobacterium bovis* (*M. bovis*) is the causative agent of bovine tuberculosis; however, it has a broader host range, including human beings. *M. bovis* has worldwide distribution and is the second most frequent etiological agent of human TB responsible for approximately 5% of the global tuberculosis burden [1]. Human tuberculosis caused by *M. bovis* and *M. tuberculosis* (*M. tb*) are clinically, radiologically, and histopathologically indistinguishable [2]. *Mycobacterium avium* subspecies *paratuberculosis* (MAP), a member of *Mycobacterium avium*, complex is the causative agent of Johne’s disease (JD) or paratuberculosis in ruminants. *Paratuberculosis* has a global distribution and is listed under the World Organization for Animal Health (OIE) Terrestrial Animal Health Code [3]. MAP infection poses a serious threat to human populations, besides affecting a wide range of animal species [4]. Contaminated food and water are the major sources of MAP infection in humans [4]. The relation between MAP and Morbus Crohn disease (CD) of humans was reported for the first time by Dalziel, in 1913 [5]. Various studies have documented the involvement of MAP in CD, but the role of MAP as an etiology of CD is debatable [4,6]. In light of the existing knowledge about MAP and its relationship to human diseases, the majority of scientists support the theory that MAP causes CD in some genetically susceptible human hosts, although additional studies are required to establish whether MAP is an etiological agent of CD [6].

The species of the mycobacterium complex persistently survive in the host mononuclear phagocytic cells, especially in the macrophages, by subverting its protective immune responses [7]. Macrophages are important mononuclear phagocytic cells playing crucial role in regulating protective immune responses for the elimination of intracellular pathogens [8]. Macrophages are key cells involved in the pathogenesis of tuberculosis and paratuberculosis by providing a niche for the persistent survival and growth of intracellular bacteria [8].

Tyrosine kinases inhibitors (TKI) are a novel class of anticancer drugs targeting cellular pathways over-expressed in various types of malignancies [9]. Nilotinib, is an important second-generation tyrosine kinase inhibitor (TKI), widely used in the treatment of chronic myeloid leukaemia (CML) [9]. It has been reported that nilotinib (Scheme 1) is the next generation of imatinib, as the first approved inhibitor of BCR-ABL, tyrosine kinase, determined the age of treatment of CML, and later studies determined its additional activity in targeting c-Kit and platelet-derived growth factor receptors (PDGFRs) [10]. Furthermore, the spectrum of BCR-ABL inhibitors has increased with the development of additional TKIs; however, the broader the spectrum of the TKI, the higher the possibility of side effects and reduction in the potential for target [10]. Growing studies reported a potential role of nilotinib in neurodegenerative diseases. Lonskaya and colleagues [11] determined that nilotinib enhanced autophagic degradation of amyloid through parkin-beclin-1 interaction in Alzheimer’s disease. Additionally, it has been studied that nilotinib induced autophagy in hepatocellular carcinoma mediated by AMP-activated protein kinase (AMPK) activation [12]. Increasing evidence suggests that nilotinib plays an important role in the regulation of autophagy and apoptosis [12,13,14] by targeting c-ABL kinase [15]. Previous reports suggested the role of c-ABL kinase in the activation of immune suppressive M2 macrophages via overexpression of peroxisome proliferator-activated receptor-gamma (PPAR-γ) [16]. Recent studies revealed that c-ABL kinase is activated in murine bone marrow derived macrophages (BMDM) infected with mycobacteria [17]. c-ABL kinase promotes the intracellular survival of *M. tb.* via inhibition of phagosomal acidification [18]. Additionally, c-ABL also contributes towards the inhibition of activated parkin protein [15,19]. Activated parkin is crucial for the degradation of accumulated toxic materials via regulation of autophagy [20]. Parkin plays an important role in the autophagic degradation of *M. tb* [21].

Autophagy is a conserved cellular process for maintaining cellular homeostasis and eliminating invading pathogens. Microtubule-associated protein 1A/1B-light chain 3 (LC3) is a key marker for monitoring autophagic flux in mammalian cells [22]. Another widely used marker for autophagic flux is the autophagy receptor sequestosome 1 (SQSTM1, p62). SQSTM1/p62 physically links the cargo to the autophagic membrane [23]. Primarily, p62 is degraded by autophagy, and the inhibition of lysosomal degradation of autophagic cargo leads to the accumulation of P62 [24]. The lysosome-associated membrane protein 1 (LAMP-1) and its family member LAMP-2 are important for the integrity of the lysosomal membrane and the fusion of the lysosome with the phagosome [25]. Previously, reports suggested that an increase in the expression or colocalization of mycobacterial phagosomes with LAMP-1 resulted in enhanced mycobacterial phagosome maturation [26]. Autophagy is regulated by multiple pathways, either dependent or independent of mTOR (mammalian target of rapamycin) [27]. It has been shown that mTOR, being a member of the canonical PI3K/Akt/mTOR pathway [28], is one of the multiple signaling pathways activated by c-ABL [29]. Recently, increasing reports have highlighted the importance of drug resistance among bacterial pathogens, including *M. tb.* [30]. Therefore, host-directed therapeutic approaches are mandatory to target those pathogens which adopting various escaping strategies to overcome host defense mechanisms [27,30]. These alternative approaches for the development of antimicrobial agents would minimize the emergence of new drug-resistant strains.

The aim our study was to evaluate the potential role of nilotinib on the development of host protective immune responses for the control of intracellular pathogenic *M. bovis* and MAP. We observed that nilotinib improved the ability of the infected macrophages for the elimination of both *M. bovis* and MAP. Nilotinib triggered autophagic degradation of intracellular mycobacterium via attenuating the signaling cascades of PI3k/Akt/mTOR pathway mediated by c-ABL. In addition, nilotinib triggered the activation of the parkin protein, which subsequently enhanced ubiquitination of *M. bovis* in infected macrophages. Nilotinib significantly restricted the survival and growth of *M. bovis* in infected mice and reduced the severity of disease. Although, we found no significant effect of nilotinib on the activation of CD4+ and CD8+ T cells; however, a noticeable reduction in the pathogenesis of *M. bovis* was observed in nilotinib treated mice. Altogether, our data reveal that nilotinib can potentially be used as a host directed therapeutic remedy for the control of intracellular *M. bovis* or MAP infection.

## 2. Materials and Methods

### 2.1. Ethics Statement

All animal experiments were performed according to the protocols for the care of laboratory animals, Ministry of Science and Technology People’s Republic of China, and approved according to animal care and use committee (IACUC) protocols with license number of 20110611-01 at the China Agricultural University, Beijing. The animal study proposal was evaluated and approved by The Laboratory Animal Ethical Committee of China Agricultural University, Beijing, China. All experiments related to bacterial culture, cells infection, and animal infection were conducted under strict biosafety measures in BSL-III facilities in National Transmissible Spongiform Encephalopathy (NTSE) laboratory, as directed by the University Institutional Biosafety Committee (IBC) approved protocols.

### 2.2. Antibodies and Reagents

Nilotinib (AMN-107) catalog number S1033 with molecular weight 529.52 was purchased from Selleckchem.com, having 99.89% purity. Rapamycin (Sirolimus; 53123-88-9) and 3metyladenine (3MA) were purchased from Selleckchem (Houston, TX, USA). PMA (Phorbol 12-myristate 13-acetate) (S1819) was purchased from Beyotime biotechnology (Beijing, China). Middlebrook, OADC (oleic acid, albumen, dextrose and catalase) (211886), and mycobactin-J, an essential component of the mycobacterial culture medium, was obtained from the Becton, Dickinson and Sparks Company (Sparks, MD, USA). The Murine Macrophage colony stimulating factor (M-CSF), (B2718) was obtained from Peprotech Technology (Wuhan, Hubei, China). The Diaminobenzidine (DAB) kit (ZLI-9017) was purchased from ZSGB-BIO sciences (Beijing, China). The calorimetric mouse IL-4, IL-10, IL-12, and IFN-γ ELISA kit (EMC003, EMC005, EMC109, and EMC101) were purchased from Biosciences. The cell titer 96 aqueous one solution cell proliferation assay kit was purchased from promega technology (G3580) (‎Fitchburg, WI, USA). The rabbit monoclonal anti-phospho-PI3 Kinase antibody (4228), rabbit anti-phospho-mTOR antibody (S2481) and rabbit anti-cABL antibody (2862T) were purchased from Cell Signaling Technology (Danvers, MA, USA). The rabbit polyclonal anti-phospho-Akt (MAP3K7IP2), rabbit polyclonal Ubiquitin antibody (10201-2-AP), rabbit polyclonal anti-Parkin antibody (14060-1-Ap), rabbit polyclonal LC-3 antibody (18725-1-AP), rabbit polyclonal anti-p62 antibody (18420-1-AP), rabbit polyclonal anti-LAMP-1 antibody (21997-1-AP), rabbit polyclonal anti-GAPDH antibody (10494-1-AP), rabbit polyclonal anti-β-actin antibody (20536-1-AP), and goat anti-rabbit IgG secondary antibody (SA000012) conjugated with peroxisome were purchased from Proteintech Biotechnology (Wuhan, Hubei, China). The donkey anti-rabbit IgG Alexa Fluor 594- conjugated secondary antibody was purchased from Yeasen Technology (34212ES60). Alexa Fluor 488 caboxylic acid succinimidyl ester (A20000) was purchased from thermofisher scietific. SiRNA-NC (SC-37007) and SiRNA-cABL (SC-29844) were purchased from Santa cruz biotechnology (Santa Cruz, CA, USA).

### 2.3. Bacterial Culture Preparation

We used two species of the Mycobacterium complex, *M. bovis* and MAP. *M. bovis* C68004 strain, which were obtained from the China Institute of Veterinary Drug Control (CVCC, China) Beijing, China. The MAP-0908 strain was isolated from infected milk of cattle and the molecular identification and characterization was done in the NTSE laboratory [31]. For the maintenance culture, *M. bovis* and MAP were grown in a Middlebrook 7H9 medium supplemented with mycobactin J (2.0 mg/L), 10% oleic-acid-dextrose-catalase (OADC), and incubated at 37 °C. For the *M. bovis* culture, the medium was additionally supplemented with sodium pyruvate at 4 mg/mL [32]. Prior to cell or animal infection, the organisms were cultured for 2 to 3 weeks to a concentration of about 10^8^ bacterial cells/mL of culture medium.

### 2.4. Preparation of Macrophages for In Vitro Experiments

We used THP-1 and murine macrophages (BMDM and RAW264.7) in the current study. The primary macrophages were derived from the bone marrow of mice and were cultured in RPMI 1640 cell culture medium as described previously [33]. In brief, BMDMs were prepared from cells obtained from femurs and tibia of six to eight-week-old C57BL/6J mice, and cultured in RPMI supplemented with 10% Fetal bovine calf serum (FBS) (Hyclone) in the presence of 10 ng/mL of M-CSF (Peprotech) and 1% Penicillin-Streptomycin. On day four, non-adherent cells were collected and cultured for three more days in the presence of fresh RPMI containing macrophage colony-stimulating factor (M-CSF) (10 ng/mL). After seven to ten days of culture, the adherent BMDM cells (6–8 × 10^6^ cells per dish) were collected and plated in 12 or 24-well cell culture plates for further experiments [34]. The RAW264.7 cell line was obtained from cold storage (−80 °C) and cultured in a cell culture flask for 24 to 48 h in DMEM medium added with 10% FBS and 1% Penicillin-Streptomycin. THP-1 cells were also obtained from cold storage and cultured in cell culture flasks in RPMI 1640 cell culture medium. THP-1 cells were stimulated with PMA (20 nM/mL) for 24 h [30]. The cells were transferred to 12 or 24 well cell culture plates for 12–18 h prior to further experiments.

### 2.5. Cells Infection

Cells were cultured in 12-well plates (2 × 10^5^ cells in each well) for 12 to 18 h to become stable. The next day, the cells were challenged with *M. bovis* and MAP with a multiplicity of infection (MOI) 1:10 (cell:bacteria) in antibiotic free cell culture medium for 3 h at 37 °C in 5% CO_2_ [35]. After incubation_,_ the supernatant was removed, and each well was washed three times with warm PBS to remove non-adherent bacilli. After washing with PBS, fresh RPMI 1640 for THP-1 and BMDM cells, while DMEM medium for RAW264.7 cells supplemented with 10% serum was added for the specified time periods. The cells were collected at different time points post-infection and samples were stored at −80 °C until further use.

### 2.6. Animal Model of Infection

BABL/C mice were obtained from specific-pathogen-free facilities at Charles River Laboratories (Catalog No: 20110428001) (Beijing, China). A total of 170, six to eight week old female BABL/C mice were used in the present study. The animals were maintained in cages under BSL3 (Biosafety level-III) conditions [32]. Mice were infected intranasally (i.n.) with 200 viable colony forming units (CFUs) of *M.bovis* or with PBS as control. Briefly, mice were anaesthetized by injecting ketamine diluted in PBS (Sigma Chemical Co., St. Louis, MI, USA) according to previous protocols [36]. Next day, five animals were randomly selected and scarified for evaluation of viable bacteria in their lungs (Appendix A). All the experimental animals were randomly divided into 4 groups: (1) no *M. bovis*-Vehicle group was treated with 30 µL DMSO (Dimethyl sulfoxide) intraperitoneal (i.p) diluted into PBS (making the final concentration of 10%) [11] as a negative control group with only vehicle, (2) the *M. bovis* + vehicle group administered with 30 µL DMSO (i.p.) as a positive control group was only administered with the vehicle on alternate days starting after one week of the bacterial challenge; (3) *M. bovis* + Nilotinib 5mg/kg injection i.p., and (4) *M. bovis* + Nilotinib 10 mg/kg injection i.p. on alternate days. The dose for nilotinib used in our current animal study was a clinically relevant dose reported by previous studies [11,15,37]. The total body weight of all experimental groups was calculated once weekly.

### 2.7. ELISA for Cytokines

The concentration of IL-4, IL-10, IL-12, and IFN-γ in the blood serum samples was measured by ELISA assay as described previously [38]. All reagents, standard, and samples were prepared according to the manufacturer’s protocols. Standards and samples (100 µL each) were added into appropriate wells of 96-well ELISA plates and incubated for 90 min at 37 °C in a humid environment. After incubation, the liquid was discarded, and the plates were washed 4–5 times with a washing buffer. Then, 100 µL of detection antibody solution was added into each well, incubated in humid conditions at 37 °C for 60 min followed by the washing steps, as previously described. Then, we added 100 µL of HRP-conjugated antibodies to each well for 30 min and incubated at 37 °C in a humid environment and repeated the washing steps. After that, we added 100 µL TMB substrate for 15 min at room temperature in the dark. After incubation, 100 µL of stop solution was added in each well to stop the reactions. To obtain the optical density, the plate was read at 450 nm by using an ELISA plate reader (Thermo Scientific Multiskan FC, USA). A standard curve was obtained by using 2-fold dilutions for calculating the concentration of cytokines 

### 2.8. Western Blot Analysis

For a Western blot assay, the total protein was collected from cells at various time points after infection. Cells were washed with PBS before being lysed with the required amount of RIPA lysis buffer (Beyotime, Beijing, China) (Tris-HCl: 50 mM, pH 7.4, NP-40: 1%, Na-deoxycholate: 0.25%, NaCl: 150 mM, EDTA: 1 mM) containing a cocktail of protease and phosphatase inhibitors [39]. Then, the samples were transferred to a centrifuge and centrifuged at 20,000× *g* for 5 min at 4 °C. The supernatants were collected and boiled for 10 min after addition of a loading buffer (250 mM Tris-HCl pH 6.8, 10% SDS, 0.5% BPB, 50% glycerol, 0.5 M DTT). Then, an equal quantity of protein from each sample was separated by using 12% or 10% SDS-PAGE electrophoresis [38,40]. After separation the proteins were transferred onto PVDF membranes (Millipore Corporation, Billerica, MA, USA) and the blots were blocked by 5% non-fat milk in TBST (25 mMTris base, 137 mM sodium chloride, 2.7 mM potassium chloride and 0.05% Tween-20, pH7.4) for 1 h at room temperature. The blots were incubated with primary antibodies overnight at 4 °C. This was followed by incubating membranes with HRP-labeled secondary antibodies for 50–60 min at 37 °C. Protein bands were obtained with ECL detection kit; images were visualized by using BIO-RAD imaging system (Bio-Rad, Hercules, CA, USA).

### 2.9. Enumeration of Viable Bacteria

For enumeration of viable bacilli, THP-1, BMDM, and RAW264.7 cells (2 × 10^5^ cells in each well) were cultured in 12-wells plates for 12 to 18 h, and then treated with nilotinib and transfected with Si-RNA [40], before infection with *M. bovis* and MAP. After incubation for a specified duration, the cells were lysed with 0.1% Triton X 100. For in-vivo experiments lung and spleen tissues were aseptically removed after sacrificing mice at different time points to enumerate viable *M. bovis* bacilli. Lung and spleen tissues were lysed with small ceramic beads in phosphate buffered saline (PBS), in a tissue homogenizer apparatus (WKT technology) in accordance with the guidelines of manufacturer [32]. For all tissue and cell samples, an appropriate tenfold dilution was prepared in PBS and transferred to Middlebrook 7H11 agar plates added with mycobacterium growth supplement such as OADC, mycobactin-J, and sodium pyrvuate in triplicate. The colonies were counted after incubation at 37 °C for 2–3 weeks for *M. bovis* and 4–5 weeks for the MAP organism.

### 2.10. Immunofluorescence and Confocal Microscopy

BMDM cells were cultured in 24 well plates containing round shaped chamber slides (Becton Dickinson) overnight. The cells were first transfected with SiRNAs or treated with nilotinib then infected with *M. bovis* for 3 h with single-cell suspensions of *M. bovis* expressing green fluorescent. After 3 h of infection the cells were washed with warm PBS thrice and incubated in fresh medium for indicated time periods. Infected BMDM cells were fixed with paraformaldehyde (4%) for 10–15 min, and then blocked with BSA for 15 min at 37 °C. After blocking, the cells were stained with mouse mAb against LC3, LAMP1, Parkin, and Ubiquitin overnight at 4 °C followed by anti-mouse Alexa Fluor 488 and DAPI [41]. Cells were analyzed by an Olympus FV1000 confocal microscope. For quantification of colocalization of various autophagy related proteins with *M. bovis* bacilli, an average of 125 infected cells were counted from three independent experiments. In addition, colocalization was measured by calculating the Manders overlap coefficient using Image-J analysis software [42].

### 2.11. Transmission Electron Microscopy

For analysis of the autophagosome formation, we conducted Transmission electron microscopy (TEM), as described previously [41]. Briefly, BMDM cells were treated with DMSO or Nilotinib and infected with *M. bovis*. After incubation, cells were washed with warm PBS and fixed in 2.5% *v*/*v* glutaraldehyde in 0.2 M phosphate buffer. Cells were again fixed in 2% osmium tetroxide and 100 mM cacodylate buffer, dehydrated with gradually higher concentrations of ethanol and progressively infiltrated with Epon resin (Pelco, Redding, CA, USA). Under various magnifications of transmission electron microscope (JEM-1230, Tokyo Japan) thin sections of each sample were examined after staining with uranyl acetate and lead citrate.

### 2.12. Cell Viability Assay

An MTS tetrazolium assay kit was used to determine the viability of cells as described previously [38]. Briefly, cells were treated with nilotinib or transfected with Si-NC and Si-cABL for the required time period. After treatment, about 20 µL of MTS reagent was added into each well followed by incubation at 37 °C for 3 h in a humidified 5% CO_2_ atmosphere. Optical density (OD) was calculated by measuring the absorbance at 490 nm wavelength by using the ELISA reader.

### 2.13. Histopathological Analysis

Gross and histological studies were carried out to evaluate the effect of nilotinib treatment on different organs of *M. bovis* infected mice. Lungs, spleens, kidneys, and livers from the sacrificed animals were collected under a sterile environment as soon as possible [32]. For gross pathology evaluation, lungs and spleens were weighed and clear images were captured at different time points of infection. For histological study, tissues were fixed in a 10% formaldehyde solution, embedded in paraffin, and cut into sections using a microtome (Leica RM2235; Leica, Buffalo Grove, IL, USA). Tissue was mounted on glass slides, deparaffinized, and stained with: haematoxylin and eosin (H&E) or Ziehl-Neelsen (ZN) for visualizing the acid-fast *M. bovis* bacilli. Sections stained with H&E or ZN were observed under low (10× or 20×) and high magnification (40× or 100×) by using an Olympus DP72 microscope fitted with a DS-Ri2 camera (Olympus, Instruments Inc., Tokyo, Japan).

### 2.14. Immunohistochemical Analysis

Lung tissue sections were processed for immunohistochemical (IHC) staining as described previously [39]. Briefly, after the deparaffinization and antigen repossession step, the endogenous peroxidase activity was blocked by dipping the slides in a 1:100 dilution of 30% hydrogen peroxide (Sigma) in a methyl alcohol solution for 15 min. After washing, the unbinding sites were blocked with BSA followed by adding LC3-II and Parkin purified rabbit anti-mouse antibodies at a concentration of 1:50 and then incubated overnight at 4 °C. After incubation, the sections were washed with PBS thrice for 5 min each, followed by incubation for 90 min at 37 °C with HRP-labeled secondary antibodies. After washing, enzymatic activity was revealed by using 3, 3′-Diaminobenzidine (DAB). Digital images were collected on an Olympus DP72 microscope fitted with DS-Ri2 camera. To quantify the intensity of LC3 and Parkin, 3 lung sections from independent animals of each group were visualized under low (40×) and high (100×) power of magnifications. The stained area compared with the total tissue area was determined by using Image-J software.

### 2.15. Flow Cytometry Analysis

Lung and spleen tissues were harvested at various time points after the challenge with *M. bovis*. The tissues were transferred to sterilized petri dishes (50 mm) containing an RPMI 1640 medium, 10% FBS, 1% penicillin/streptomycin. Single-cell suspensions from lung and spleen tissue were obtained as described previously [43]. Single-cell suspensions were stimulated with Early secretory antigenic target-6 (ESAT-6) (5 μg/mL) for 12–18 h at 37 °C. Cell suspensions were then stained with fluorochrome labeled antibodies for anti-CD4 (anti-mouse CD4 FITC), anti-CD8 (Anti-Mouse CD8a APC), and anti-CD3 (Anti-Mouse CD3e PerCP-Cyanine5.5) from eBioscience (San Diego, CA, USA). Cells were analyzed with BD FACSVerse^TM^ flow cytometer (BD Biosciences, San Jose, CA, USA) using FlowJo X software.

### 2.16. Statistical Analysis

All the cell experiments were performed on three separate occasions. Data are expressed as means ± SD. The student *t*-test was used for comparison between two experimental groups. One way or two-way ANOVA followed by Tukey’s multiple comparison and Bonferroni’s multiple comparison tests was performed for multiple-group comparisons. Statistical analysis was carried out by using GraphPad Prism v5 software (GraphPad, La Jolla, CA, USA). *p* values < 0.05 were considered statistically significant.

## 3. Results

### 3.1. Nilotinib Inhibits the Intracellular Survival of M. bovis and MAP In Vitro

Nilotinib plays an important role in modulating signaling pathways leading for the induction of autophagy in various tumor cells [12,13]). We assessed whether nilotinib might function in modulating protective immune responses for the control of intracellular mycobacterium. BMDM and RAW264.7 cells were treated with various concentrations of nilotinib (5–40 µM/mL) 2 h prior to *M. bovis* and MAP infection. We observed an increased level of LC3-II protein after 24 h of *M. bovis* (Appendix A) and MAP infection (Appendix A) in a dose dependent manner compared to the control group. In addition, we found a gradual reduction in P62 protein level in macrophages treated with nilotinib. A cell viability assay revealed that a concentration of 30 µM and above of nilotinib significantly affected the viability of macrophages (Appendix A). Therefore, we treated the cells with 10 µM and 20 µM to evaluate the effect of nilotinib on the growth and survival of *M. bovis* and MAP. The concentration of nilotinib (10 µM and 20 µM) we used in this study for in vitro experiment was a clinically relevant dose reported by Fei and colleagues [44]. In addition, no significant effect of nilotinib (10 µM and 20 µM) was observed on the phagocytic ability of macrophages (Appendix A). CFU results revealed that nilotinib significantly reduced the viable bacterial count of both *M. bovis* and MAP at 48 and 72 h post treatment in BMDM, RAW264.7, and THP-1 cells (Figure 1A–F). These results suggest that nilotinib restricted intracellular mycobacterial growth in macrophages.

### 3.2. Nilotinib Induces Autophagy during M. bovis Infection

We further investigated the role of nilotinib on the regulation of autophagy in macrophages after infection with *M. bovis*. As shown in Figure 2, the expression of LC3-II and LAMP-1 increased in nilotinib treated cells in comparison to control group upon *M. bovis* infection, while a gradually decreased level of p62 was found (Figure 2A,B). In addition, we also observed that rapamycin (an autophagy inducer) enhanced the effect of nilotinib on LC3-II and P62 expression, while 3MA (an autophagy inhibitor) altered LC3 lipidation induced by nilotinib (Figure 2C,D). The immunofluorescence analysis showed that the colocalization% of LC3-II and LAMP-1 with *M. bovis* in BMDM cells wasenhanced upon treatment with nilotinib as compared to untreated control cells (Figure 2E,F). To further clarify the induction of autophagy upon nilotinib treatment in *M. bovis* infected cells, we determined autophagosomes in cross-sections of treated and untreated BMDM cells, after the challenge with *M. bovis*, by transmission electron microscopy (TEM). TEM results revealed that nilotinib treated cells showed higher numbers of autophagosomes per cellular cross-section compared to untreated controls in *M. bovis* infected cells (Figure 2G). Similarly, IHC results showed increased immunostaining for LC3 proteins in the lung sections of nilotinib treated mice upon *M. bovis* infection compared to untreated infected mice (Figure 2H). Collectively, these results demonstrated that nilotinib induced autophagy during *M. bovis* infection.

### 3.3. Nilotinib Attenuates c-ABL Dependent PI3k/Akt/mTOR Signaling Pathway in M. bovis Infected Macrophages

It has been reported that the PI3k/Akt/mTOR signaling pathway is important for multiple cellular functions including cell survival [45] and autophagy [46]. We evaluated the effect of nilotinib on signaling cascades of the PI3k/Akt/mTOR pathway in *M. bovis* infected macrophages. As shown in Figure 3, nilotinib significantly inhibited the phosphorylation of PI3k, Akt, and mTOR in macrophages infected with *M. bovis* compared to the untreated control group (Figure 3A,B). Previous reports suggested that nilotinib blocks the activation of c-ABL [47,48], while c-ABL activates PI3k/Akt/mTOR pathway [28]. Furthermore, Tyrosine kinase c-ABL plays an active role in bacterial infections including *M. tb* [18,49]. We observed a time dependent increase of c-ABL in both BMDM and RAW264.7 cells infected with *M. bovis* (Appendix A). Nilotinib significantly inhibited the activation of c-ABL in *M. bovis* infected macrophages as compared to untreated controls (Figure 3C,D). We further investigated whether nilotinib mediated activation of PI3k/Akt/mTOR pathway; therefore c-ABL expression was interfered by transfecting BMDM cells with Si-cABL (Figure 3E). In addition, Si-cABL transfection showed no significant effect on the viability and phagocytic ability of macrophages (Appendix A). As illustrated in Figure 3F, the abrogation of c-ABL enhanced the effect of nilotinib on the phosphorylation of PI3k, Akt, and mTOR in macrophages infected with *M. bovis* as compared to Si-NC (Figure 3F). Additionally, Si-cABL also contributed towards the effect of nilotinib in the lipidation of LC3 and reduced the level of p62 in infected macrophages (Figure 3G). Immunofluorescence results showed that silencing of c-ABL significantly enhanced the effect of nilotinib on the colocalization of LC3-II with *M. bovis* (Figure 3H). CFU results revealed that transient attenuation of c-ABL contributed towards the autophagic degradation of *M. bovis* upon nilotinib treatment (Figure 3I). These results suggest that nilotinib promoted autophagy in *M. bovis* infected macrophages by inhibiting the c-ABL mediated PI3k/Akt/mTOR pathway.

### 3.4. Nilotinib Increases Parkin Activation During M. bovis Infection

Parkin is a ubiquitin E3 ligase essential for elimination of pathogenic *M. tb.* [21]. We investigated whether nilotinib participates in the activation of parkin proteins in macrophages upon infection with *M. bovis*. A noticeable increase in parkin protein level was observed in nilotinib treated cells in comparison to untreated control cells (Figure 4A,B). Immunofluorescence results showed a significant increase in the colocalization of parkin with *M. bovis* in nilotinib treated cells (Figure 4C). In addition, IHC results from lung sections of infected mice revealed increased signals of parkin protein upon nilotinib treatment (Figure 4D). Several independent reports found that c-ABL inhibits parkin’s E3 ubiquitin ligase activity [50]. As mentioned in Figure 3, the attenuation of c-ABL expression (Figure 3E) significantly enhanced the effect of nilotinib on the activation of parkin protein in *M. bovis* infected macrophages (Figure 4E). Moreover, an increased colocalization of parkin with *M. bovis* was observed in macrophages transfected with Si-cABL followed by nilotinib treatment compared to Si-NC untreated cells (Figure 4F). These findings revealed that nilotinib modulated parkin activation both in-vitro and in-vivo during *M. bovis* infection and contributed towards reducing the severity of infection.

### 3.5. Nilotinib Promotes Ubiquitination of M. bovis in Infected Macrophages

It has been demonstrated that parkin and its E3 ligase activity are important for the colocalization of ubiquitin with *M. tb* [21]. We hypothesized that nilotinib may recruit ubiquitin upon *M. bovis* infection. Indeed, after infection of BMDM and RAW264.7 cells with *M. bovis,* we found significantly abundant poly-ubiquitin chains in nilotinib treated cells in comparison to the untreated controls (Figure 5A,B). Additionally, we found a high% colocalization of ubiquitin with alexa flour expressing *M. bvois* in nilotinib treated cells in comparison to the untreated controls (Figure 5C). The silencing of c-ABL further enhanced the effect of nilotinib on the intensity of poly-ubiquitin chain (Figure 5D). Similarly, the effect of nilotinib on the colocalization % of ubiquitin with stained *M. bovis* was further enhanced by Si-cABL in macrophages compared to untreated Si-NC cells (Figure 5E). Altogether, these results suggested that c-ABL inhibition is essential to modulate parkin mediated ubiquitination of *M. bovis* in murine macrophages.

### 3.6. Nilotinib Reduces the Severity of M. bovis Pathogenesis in Mice

Our cell experiments revealed that nilotinib promoted the bactericidal ability of macrophages to eliminate intracellular mycobacterium via the induction of autophagy. Here, we infected mice with pathogenic *M. bovis* and after one week of infection, nilotinib was injected at a dose rate of 5 mg/kg and 10 mg/kg on alternate days. The dose of nilotinib used in the current study for in-vivo experiments was under a clinically relevant dose as previously reported [15]. Animals were slaughtered at various time points after infection (Figure 6A). The total body weight of animals and their lung and spleen weights revealed that nilotinib suppressed the degree of pathogenesis of *M. bovis* (Appendix A). Gross analysis showed that nilotinib reduced the development of lesions in the lungs of infected mice as compared to the untreated control group (Figure 6B). At an early stage of infection (35 dpi), there was no significant difference in the spleen size, while in the later stages untreated mice (images labeled with II) showed enlarged spleens compared to those in the treated infected mice (images labeled with III and IV) (Figure 6C). Furthermore, the lung lobe showed a significantly high% area covered by lesions in untreated mice compared to treated *M. bovis* infected mice at all time points (Figure 6D,E). In addition, the increased number of bacilli was observed in the lung sections of untreated animals in comparison to treated animals after staining with the Ziehl-Neelsen staining method (Appendix A). Next, the harvested lung and spleen tissues were subjected to histopathological analyses. At early stages of infection (35 dpi), treated mice had small foci of inflammation composed of epithelioid macrophages and lymphocytes with no clear evidence of necrosis (Data not shown). However, at a later stage of infection (63 dpi), at high magnification, granulomatous lesions from untreated animals showed necrotic areas (marked with red arrows) in the central region of the lesion, which represent the severity of the disease (Figure 6F). Interestingly, lung sections from nilotinib treated animals showed comparatively less severe lesions with a reduced necrotic core (Figure 6G,H). Notably, no histological changes were observed in the H&E stained sections of the livers and the kidneys of nilotinib treated mice compared to the untreated controls (Appendix A). These findings suggested that nilotinib contributed towards minimizing the severity of *M. bovis* pathogenesis in mice.

### 3.7. Nilotinib Reduces M. bovis Burden Irrespective of Modulating T-Cell Function

Although antigen-specific CD4 T cells and IFN-γ are important for controlling *M. tb* infection, they are not sufficient for eliminating the pathogen [51]. We established the protective role of nilotinib on the development of immune responses against *M. bovis* infection. To analyze the effects of nilotinib on T-cells-mediated immune responses against *M. bovis*, we determined whether nilotinib modulates CD4 and CD8 T-cells associated functions. Flow cytometry analysis showed that nilotinib treatment has no significant effect on the proliferation of CD4 and CD8 T-cells in both the lung (Figure 7A–C) and spleen (Figure 7D–F) tissues of the infected mice. Similarly, no significant difference was observed on the secretion of cytokines (IFN-γ, IL-4, IL-10 and IL-12) between treated and untreated animals (Figure 7G–J).

As illustrated, gross morphological and histological analysis of *M. bovis* infected mice showed a clear reduction in the severity of disease after treatment with nilotinib. In addition, lung sections stained with the Ziehl-Neelsen stain revealed a reduced number of bacilli after nilotinib treatment (Appendix A). To further understand the prognosis of *M. bovis* infection in nilotinib treated animals, lung and spleen tissues were used for enumeration of viable bacilli. A noticeable reduction was observed in viable bacilli count in lung and spleen tissues of nilotinib treated mice at all time points after infection as compared to the untreated control group (Figure 8A,B). In addition, no mortality was observed in nilotinib treated animals in comparison to untreated infected animal. The animal survival data revealed that nilotinib treatment prolonged the survival of infected animals (Figure 8C). These results suggest that nilotinib suppressed the growth of virulent *M. bovis* in infected mice regardless of T-cell proliferation and migration but were more dependent on autophagy induction. Based on our current observations, we propose a model for the mechanism of nilotinib on the regulation of protective immune response against *M. bovis* infection. However, after binding to the Pattern recognition receptors (PRRs) of phagocytic cells, *M. bovis* inhibits the anti-mycobacterial ability of macrophages by activation of various signaling pathways. We observed that *M. bovis* leads to the activation of the PI3K/AKT/mTOR axis mediated by the over expression of c-ABL, which impairs the assembly of autophagosome and inhibit the degradation of intracellular pathogen. In addition, c-ABL also inhibited parkin mediated ubiquitination of *M. bovis* bacilli in macrophages. Nilotinib induced autophagic degradation of intracellular mycobacterium that occurred via the inhibition of the PI3k/Akt/mTOR axis mediated by c-ABL. Furthermore, nilotinib promoted ubiquitin accumulation around *M. bovis* through activation of the E3 ubiquitin ligase parkin (Figure 8D). Altogether, these findings illustrated the significance of nilotinib in the regulation of innate immune responses, which control the multiplication of intracellular mycobacterium.

## 4. Discussion

Autophagy is a highly conserved catabolic process playing a vital role in various cellular pathways, ranging from cell growth and differentiation to aging and cell death [52]. Xenophagy, a specialized form of autophagy, has been described as a primary defense strategy against certain intracellular bacteria, including *M. tb* [26]. Autophagy has been studied extensively as a therapeutic strategy; however, either too little or too much autophagy can be deleterious for cell survival and functions [53]. Therefore, a tight regulation of the induction and magnitude of autophagy is necessary to control the growth of intracellular bacteria. Emerging evidence suggests that tyrosine kinase inhibitors play an important role in the regulation of autophagy associated signaling pathways [12,54]. We found that nilotinib promotes the anti-mycobacterial ability of macrophages via induction of autophagy. It has been shown that *M. tb* phagosomal membranes labeled with lysosomal associated membrane proteins 1 (LAMP-1) are associated with lysosome phagosome fusion, which leads to the degradation of intracellular *M. tb* [55]. In the current study, we observed that nilotinib increased the colocalization of LAMP-1 with *M. bovis* that leads to the fusion of lysosome with the phagosome. Our findings are in line with those of Gutierrez et al. [26], demonstrating that the presence of LAMP-1 showed the association of lysosome phagosome fusion and the progression of the autophagic degradation of intracellular bacteria. Previous studies demonstrated that the Src tyrosine kinase inhibitor significantly reduced the survival of virulent *M. tb* in macrophages [30]. Similarly, we achieved a considerable reduction in the intracellular load of *M. bovis* and MAP in infected macrophages after nilotinib treatment.

Tyrosine kinase c-ABL regulates several cell functions including cell survival and multiplication [18]. Previous reports suggested that c-ABL prevent phagosomal acidification to enhance *M. tb* growth and survival [18]. Therefore, we investigated the role of nilotinib in the involvement of c-ABL activity on the modulation of immune responses against *M. bovis* infection. Mahadik and colleagues observed that imitinib inhibited c-ABL in *M. tb* infected RAW264.7 cells [40]. We observed a significant inhibition of c-ABL activity in *M. bovis* infected macrophages treated with nilotinib. It is reported that c-ABL is one of the key regulators of various signaling pathways in host cells, including the mTOR pathway [28,40]. In addition, mTOR is a master regulator of multiple signaling pathways [27], including PI3K/AKT/mTOR pathway [28]. We investigated the effect of nilotinib on the PI3k/Akt/mTOR pathway in *M. bovis* infected macrophages. In accordance with the findings of Airiau et al., we observed that nilotinib reduced the phosphorylation of PI3k, Akt, and mTOR in *M. bovis* infected macrophages [28]. Similarly, Chandra and colleagues reported that the inhibition of Src tyrosine kinase in THP-1 cells reduced Akt phosphorylation and increased *M. tb.* autophagic degradation [30]. Furthermore, c-ABL silencing enhanced nilotinib efficacy in the regulation of autophagy via inhibition of PI3k/Akt/mTOR signaling cascades in BMDM cells after infection with *M. bovis*. Previously it was demonstrated that the inhibition of c-ABL activation through imitinib treatment potentiated innate immune responses to limit *M. tb.* multiplication [18,40]. Stanley and colleagues reported that Abl and Akt are important for mycobacterial replication in murine macrophages [56]. Our observations are in agreement with these studies, suggesting that nilotinib restores the regulation of autophagy in *M. bovis* infected macrophages by inhibiting the PI3k/Akt/mTOR signaling pathway.

Increasing evidence suggests that parkin mediates the autophagic degradation of damaged intracellular organelles, including dysfunctional mitochondria [57,58]. Furthermore, several studies discovered the role of parkin in the induction of xenophagy for the degradation of intracellular *M. tb.* [21,59]. Hence, we reasoned that parkin via its E3-ligase activity might affect *M. bovis* survival and multiplication in macrophages. We found that nilotinib enhanced the recruitment of parkin to *M. bovis* in BMDM cells. Several other E3 ubiquitin ligases, such as NEDD4 and Smurf1, represent important players in autophagy by mediating the direct ubiquitination of *M. tb* in infected THP-1 cells [59,60]. Similarly, we found a high% of *M. bovis*-ubiquitin colocalization in nilotinib treated BMDM cells. As described previously, c-ABL had an inhibitory effect on the activation of parkin. Therefore, we asked whether nilotinib modulated parkin activation via the regulation of c-ABL. We observed that the silencing of c-ABL enhanced parkin activation and increased *M. bovis*-ubiquitin colocalization% after nilotinib treatment. Hence, we showed that nilotinib plays a central role in the modulation of innate immune signaling via activation of parkin during *M. bovis* infection.

Previous studies reported that nilotinib was capable of inhibiting both chloroquine resistant and sensitive strains of *Plasmodium falciparum*. In addition, nilotinib treatment also protected mice against plasmodium infection [61]. Furthermore, Ishiyama and colleagues suggest the possible existence of malaria kinases having a binding site structurally similar to that of human ABL kinase, a direct target of nilotinib [61]. It has been reported that bacteria have evolved tyrosine kinases; however eukaryotic-like tyrosine kinases have not been identified in bacterial genomes [62]. Additionally, it has been reported that *M. tuberculosis* protein tyrosine kinase does not possess any signature or pattern related to bacterial tyrosien-kinases and eukaryotic tyrosine kinases [63]. Increasing evidence suggests that a potential use of tyrosine kinase inhibitors is to target the host defense mechanism. Similarly, Napier and colleagues investigated the prophylactic role of imitinib in a mouse model of tuberculosis by targeting host-cell mediated immune systems [49]. Here, we investigated the mechanism and the potential of nilotinib as a therapeutic agent in *M. bovis* infection at two different dose rates (5 mg/kg and 10 mg/kg) on alternate days. A considerable difference was observed in nilotinib treated and untreated mice, despite the concentration of the drug at the level of overall lung and spleen morphology, histopathology, and weight of lungs and spleen. Previous reports suggested that nilotinib (25 mg/kg) abolished acetaminophen induced liver injury and necro-inflammation in mice [37]. We found no toxic effects of nilotinib treatment in the liver and kidney of *M. bovis* infected mice (Appendix A).

Napier and colleagues reported that imitinib treatment rescued *M. marinum* infection in mice by augmenting myelopoiesis but not lymphopoiesis [64]. It has been reported that though antigen-specific CD4 T cells and IFN-γ are important for controlling *M. tb* infection, they are not sufficient for protecting against the infection [51]. In the current study, we also found no significant difference in the activated CD4 and CD8 T cells in nilotinib treated mice after a challenge with *M. bovis*. These results are in line with the work by Varda-Bloom and colleagues showing that T-lymphocytes subsets were stable during nilotinib treatment in CML patients [65]. Similar to Mahadik et al. [40], we found a striking difference in the bacterial load in the lung and spleen tissues of nilotinib treated mice at all time points after infection compared to those in the untreated control animals. In addition, no clear difference was observed in the secretion of key cytokine in the treated and untreated mice. Our findings are in agreement with Wehrstedt et al. [66], who used a dual Abl/Src TKI dasatinib, which restricted the growth of intracellular *M. tb*.

## 5. Conclusions

We uncovered a novel role of nilotinib in the control of intracellular bacterial infections by induction of autophagy via modulation of key autophagy regulating signaling pathways. Our study not only unraveled critical host defense mechanisms against a deadly pathogen but also deepened our understanding of the mechanisms underlying better protection and safety profiles of clinically advanced therapeutic candidates against intracellular mycobacterial species. The current study showing that nilotinib promotes antimicrobial activity in *M. bovis* infected mice without mediating inflammatory effects suggests one promising strategy for maintaining innate antimicrobial activity in a host receiving treatment for chronic inflammatory diseases.

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
