# Peer review of "Nilotinib: A Tyrosine Kinase Inhibitor Mediates Resistance to Intracellular Mycobacterium Via Regulating Autophagy"

_cells, 2019, doi:10.3390/cells8050506_

Round 1
Reviewer 1 Report
In this manuscript “Nilotinib: a tyrosine kinase inhibitor mediates resistance to intracellular Mycobacterium via regulating autophagy” authors showed that Nilotinib, an anticancer tyrosine kinase inhibiting drug, effective against M bovis infected mice. Further, they studied its mechanism of action by western blot assays and in vivo studies.
After careful reading, the reviewer has made the following comments.
The introduction section doesn’t cover nilotinib know targets, LC-3 and P62 proteins role in autophagy.
Nilotinib inhibits c-ABL tyrosine kinase as well as KIT and PDGF-R kinases. The study conducted showed that the observed antibacterial activity is because of c-ABL inhibition, but the effects of KIT and PDGF-R inhibition by nilotinib are not investigated.
Nilotinib is next generation drug of imatinib, and its shown that imatinib inhibits Mtb by regulating phagosomal acidification, the same mode of action might be possible for nilotinib in addition to c-ABL, KIT and PDGFR inhibition. This mode of action should be investigated by the authors.
Nilotinib MIC/IC50 against M bovis should be determined to show that observed effect is only because of c-ABL/KIT/PDGFR inhibition/phagosomal acidification and nilotinib doesn’t inhibit M bovis.
Why observed % colocalization is less in RAW264.7 cells?
Fig. S5B, why body weight is less for 10mg/kg nilotinib treated mice as compared to 5mg/kg?
Fig 8C is difficult to follow.
The resolution of most of the figures is poor it should be improved.
The purity of nilotinib purchased from the vendor is not mentioned.
Structure of nilotinib should be embedded in the introduction.
% of DMSO should be mentioned in the figures caption.
As western blot is important expt. of the study, the detailed procedure should be provided in addition to citing back reference.
Captions are difficult to follow, A, B, C etc. should be written in the beginning of the sentences.
Wavelength/region is not given in microscopic assays.
Despite the fact that some meaningful results were presented, the MS will suitable for publication after revising according to above-mentioned points.
The references are not presented in the same style.
Author Response
Thanks for your encouraging comments.
We incorporated all the changes properly according to the suggestions. We also reviewed the whole manuscript several times carefully to describe the findings in the text properly and checked spelling, typing and grammatical mistakes. The final version of manuscript is also checked by a native English-speaking professional person. The changes we made in the revised version of manuscript are highlighted with red color. We hope that the revised version of manuscript will be free from any mistakes but if any changes that you might suggest further, we are welcomed.
Point by Point response to the Comments and Suggestions of Reviewer 1
In this manuscript “Nilotinib: a tyrosine kinase inhibitor mediates resistance to intracellular Mycobacterium via regulating autophagy” authors showed that Nilotinib, an anticancer tyrosine kinase inhibiting drug, effective against M bovis infected mice. Further, they studied its mechanism of action by western blot assays and in vivo studies.
The introduction section doesn’t cover nilotinib know targets, LC-3 and P62 proteins role in autophagy.
Response:
Thanks for suggestion. In the introduction section we mentioned the role of nilotinib on the signaling pathways regulating autophagy. We incorporated some additional references in line number 56-66 regarding the role of nilotinib in additional targets and the induction of autophagy. We also discussed in the introduction section that LC-3 and P62 are the key markers used for the measurement of autophagy induction in line number 76-87. Our results suggest that nilotinib induced autophagy by targeting PI3K/AKT/mTOR signaling pathway via inhibition of c-ABL. In addition, PI3K/AKT/mTOR pathway negatively regulates autophagy, while we found enhanced LC-3 lipidation and reduced P62 expression which shows that nilotinib induced autophagy via attenuation of PI3K/AKT/mTOR. Collectively, our results suggest that nilotinib modulate PI3K/AKT/mTOR mediated by c-ABL for the regulation of autophagy.
Nilotinib inhibits c-ABL tyrosine kinase as well as KIT and PDGF-R kinases. The study conducted showed that the observed antibacterial activity is because of c-ABL inhibition, but the effects of KIT and PDGF-R inhibition by nilotinib are not investigated.
Response:
Thanks for comment. It has been investigated that nilotinib has several targets such as c-KIT, PDGFR and c-ABL but in the current study we determined the effect of nilotinib on c-ABL kinase function during M. bovis infection. Jiao and colleagues reported that several tyrosine kinase inhibitors target c-KIT and PDGFR (Jiao et al., 2018). However, increasing evidence suggest that nilotinib play a key role in inhibition of c-ABL. Nilotinib is more efficient than imatinib to inhibit the BCR-ABL activity. In addition, the concentration of nilotinib required to block the BCR-ABL activity, is not sufficient to inhibit c-KIT and PDGFR (Weisberg et al., 2005). The study of Stanly et al, showed that gefitinib, an inhibitor of EGFR suppressed M. tb replication in macrophages via inhibition of p38 MAPK signaling pathway (Stanely et al., 2004). It has been reported that tyrosine kinase c-KIT plays an important role in angiogenesis (Feng et al., 2015). However, studies revealed that c-ABL kinase is activated in murine macrophages infected with mycobacteria (Mahadik et al., 2018). In addition, c-ABL kinase promotes intracellular survival of M.tb via inhibition of phagosomal acidification (Bruns et al., 2012). Therefore, in the current study we investigated the role of nilotinib in the regulation of autophagy mediated by PI3K/AKT/mTOR pathway dependent on c-ABL expression during M. bovis infection. In our next project, we will explore the role of key tyrosine kinase inhibitors in comparison with nilotinib on the regulation of c-KIT and PDGER during mycobacterial infection.
Nilotinib is next generation drug of imatinib, and its shown that imatinib inhibits Mtb by regulating phagosomal acidification, the same mode of action might be possible for nilotinib in addition to c-ABL, KIT and PDGFR inhibition. This mode of action should be investigated by the authors.
Response:
We are agreed, that nilotinib is the next generation of imatinib and increasing evidence suggest that nilotinib is more effective than imatinib in BCR-ABL activity (Weisberg et al., 2005; Yu et al., 2013). It has been reported that imatinib limited M.tb multiplication in infected macrophages via regulation of phagosomal acidification. Bruns and colleagues determined the phagosomal acidification by detecting ph, colocalization of M.tb-LysoSensor and also measured cathepsin-D to determine phagosome lysosome fusion after imatinib treatment in M.tb infected macrophages (Bruns et al., 2012). However, we investigated the regulation and maturation of autophagosome by measuring key markers such as LC3 lipidation, P62 and LAMP-1 expression upon nilotinib treatment in M. bovis infected macrophages. Similar to phagosomal acidififation, M.tb phagosomal membranes labeled with lysosomal associated membrane proteins 1 (LAMP-1) is associated with the lysosome phagosome fusion which leads the degradation of intracellular M.tb (Van der Wel et al., 2007). In our current study, we measure the colocalization of LAMP-1 with M. bovis, one of key marker for the detection of lysosome phagosome fusion. As described in line number 535-540, we found an increased colocalization of LAMP-1 with M. bovis upon nilotinib treatment. Our findings are in agreement with Gutierrez et al, that the presence of LAMP-1 showed the association of lysosome phagosome fusion and the progression of autophagic degradation of intracellular bacteria (Gutierrez et al., 2005). Although, both the detection of phagosomal acidification or phagosome lysosome fusion (LAMP-1 expression) shows the maturation of autophagosome and the degradation of intracellular pathogens. As mentioned above that in our next project, we will investigate certain key TKI in comparison to nilotinib. In addition, we will also investigate additional methods for assessment of phagosomal acidification for mycobacterial phagosomal maturation after treatment with tyrosine kinase inhibitors.
Nilotinib MIC/IC50 against M bovis should be determined to show that observed effect is only because of c-ABL/KIT/PDGFR inhibition/phagosomal acidification and nilotinib doesn’t inhibit M bovis.
Response: Thanks for your comment
The tyrosine kinase inhibitor nilotinib we used in the current study, target only eukaryotic tyrosine kinases as described in line 576-581. Previous studies reported that bacteria have evolved tyrosine kinases; however eukaryotic-like tyrosine kinases have not been identified in bacterial genome yet (Grangeasse et al., 2012). Additionally, it has been investigated that Mycobacterium tuberculosis protein tyrosine kinase does not possess any signature or pattern related to bacterial tyrosine-kinases and eukaryotic tyrosine kinases (Bach et al., 2009). Bruns and colleagues also showed that imatinib has no direct effect on the viability of M.tb after incubation of extracellular bacilli with imitinib (Bruns et al., 2012). Furthermore, the main objective of our study was to find new host-directed therapeutic regimes that could also overcome the emerging problem of drug resistant strains of mycobacterium. As reported that nilotinib target eukaryotic tyrosine kinas, while bacteria have no target for nilotinib. Therefore, in the current study, we investigated the effect of nilotinib on host-immune mechanism against mycobacterial infection by targeting host tyrosine kinase. Our data shown that nilotinib enhanced macrophage antibacterial ability via regulation of autophagy. In mouse model of M. bovis infection, we observed that nilotinib reduced the multiplication of M. bovis in lung and spleen tissue of mice.
Why observed % colocalization is less in RAW264.7 cells?
Response:
Thanks, for your comment but i think maybe you got confused in our figures, as we mostly used BMDM and RAW264.7 cells in our current study, but for confocal microscopy we used only BMDM cells to analyze the % colocalization of proteins like LC-3, LAMP-1, Parkin and ubiquitin with M. bovis. In addition, we found no significant difference in the expression of various proteins observed by WB method between BMDM and RAW264.7 cells in response to various treatments.
Fig. S5B, why body weight is less for 10mg/kg nilotinib treated mice as compared to 5mg/kg?
Response: Thanks for comment. As described that no significant difference was observed in the lungs or spleen of mice treated with nilotinib 5mg/kg and 10mg/kg after M. bovis infection. In addition, no toxicity was found in the histological analysis of liver and kidneys of mice treated with nilotinib as compared with untreated mice. However, the difference in the body weight in mice treated with 5mg/kg and 10mg/kg is significant in comparison to untreated infected mice. In addition, no significant difference was observed in the body weight between mice treated with 5mg/kg and 10mg/kg, while the slight reduction in the body weight of mice treated with 10mg/kg might be the initial stress due to difference in dose rate.
Fig 8C is difficult to follow.
Response:
Figure 8 panel C represent the survival of animals of all experimental groups. As describe in the result section that no mortality was observed in both treated group until the last time point of infection. In the survival graph the black line represent uninfected control, red line represents M. bovis infected with vehicle, green line represents M. bovis infected treated with nilotinib 5mg/kg and blue line shows M. bovis infected treated with nilotinib 10mg/kg respectively. As there was no difference in control (showed with black line) and two treated groups (showed with green and blue line) therefore only blue is visible on the survival graph even in the early time point the red line couldn’t see but after 70 dpi when few animals died in the untreated infected group then the red line is visible. We also clearly described the detail in the text in line number 486-488.
The resolution of most of the figures is poor it should be improved.
Response:
Thanks for valuable suggestion, we further increased the resolution of each figure that improved the visibility of all figures.
The purity of nilotinib purchased from the vendor is not mentioned.
Response:
Thanks, the nilotinib (AMN-107) catalog number S1033 with molecular weight 529.52 was purchased from Selleckchem.com having 99.89% purity. Now we mentioned these information’s regarding nilotinib in the reagent section of the Materials and methods in line number 113.
Structure of nilotinib should be embedded in the introduction.
Response:
Thanks, the chemical structure of nilotinib has been embedded in the introduction section as suggested.
% of DMSO should be mentioned in the figures caption.
Response:
Thanks, we used 0.1% of DMSO in the controlled groups and now mentioned in the figure captions.
As western blot is important expt. of the study, the detailed procedure should be provided in addition to citing back reference.
Response:
Thanks, we described the western blot procedure in detail with reference in the materials and methods section as suggested. We also improved other sections of the materials and methods by adding detail descriptions of the procedures.
Captions are difficult to follow, A, B, C etc. should be written in the beginning of the sentences.
Response:
Thanks, we re-wrote the captions and correctly mentioned A,B,C etc, in all figures legends in the revised version of our manuscript.
Wavelength/region is not given in microscopic assays.
Response:
Thanks, in all microscopic assay, we incorporated the wavelength used for capturing images and also mentioned scale bar at the lower corner of each panel of microscopic image. In addition, we also mentioned the scale bare in the figure captions with respect to the panel.
Despite the fact that some meaningful results were presented, the MS will suitable for publication after revising according to above-mentioned points.
Response:
We incorporated all the changes properly according to the suggestions. We also reviewed the whole manuscript several times carefully to describe the findings in the text properly and checked spelling, typing and grammatical mistakes. The final version of manuscript is also checked by a native English-speaking professional person. The changes we made in the revised version of manuscript are highlighted with red colour. We hope that the revised version of manuscript will be free from any mistakes but if any changes that you might suggest further, we are welcomed.
The references are not presented in the same style.
Response:
Thanks, the references are now modified according to the journal format.

Reviewer 2 Report
Manuscript entitled “Nilotinib: a tyrosine kinase inhibitor mediates resistance to intracellular Mycobacterium via regulating autophagy” by Hussain et al. report the use of nilotinib for the treatment of M. bovis and MAP infection. Nilotinib is a second generation tyrosine kinase receptor inhibitors and used classically in the treatment of CML. Due to involvement of c-ABL a tyrosine kinase and mTOR signaling pathways with the nilotinib which triggered autophagic degradation of intracellular mycobacterium via attenuating the PI3K/Akt/mTOR pathways. Author performed several experiments including invivo to show the attenuation of PI3K/Akt/mTOR signaling pathways. Controls were used appropriately in the experiments. Activation of parkin protein was found after treatment with nilotinib which enhance ubiquitination of M. bovis and hence affect the survival of bacteria growth in the mice studies. Overall a good studies to show the use of nilotinib for treatment of intracellular bacterial infection such as MAP. Therefore, these studies are interest of broader community and manuscript could be recommended for publication after minor corrections of comments as follow.
1. line number 160: Add strain number/catalogue number of BABL/C mice purchase from Charles river lab.
2. Several typos found throughout the manuscript for the use of symbol. Author should correct them. For example: ul need to change to µL (line number 167); symbol for degree (temperature) need to correct.
2. line number :160, add sexes of mice used in the studies.
3. How nilotinib were solubilize for experiments? What was the concentration of DMSO treatment in the experiments? Just DMSO label could be read as 100% DMSO which could be toxic.
4. References need to format as per guidelines of journal.
Author Response
We are thankful for the encouraging comments/suggestions regarding to our manuscript. We hope that after addressing all these comments and positive criticisms one by one it will improve the understanding and quality of our manuscript for the general readers and scientific community actively involved in intracellular pathogen studies. We incorporated all the changes as suggested and we would surely welcome any further changes that can make our study more readable before final publication.
Point by Point response to the Comments and Suggestions of Reviewer 2
Manuscript entitled “Nilotinib: a tyrosine kinase inhibitor mediates resistance to intracellular Mycobacterium via regulating autophagy” by Hussain et al. report the use of nilotinib for the treatment of M. bovis and MAP infection. Nilotinib is a second generation tyrosine kinase receptor inhibitors and used classically in the treatment of CML. Due to involvement of c-ABL a tyrosine kinase and mTOR signaling pathways with the nilotinib which triggered autophagic degradation of intracellular mycobacterium via attenuating the PI3K/Akt/mTOR pathways. Author performed several experiments including invivo to show the attenuation of PI3K/Akt/mTOR signaling pathways. Controls were used appropriately in the experiments. Activation of parkin protein was found after treatment with nilotinib which enhance ubiquitination of M. bovis and hence affect the survival of bacteria growth in the mice studies. Overall a good studies to show the use of nilotinib for treatment of intracellular bacterial infection such as MAP. Therefore, these studies are interest of broader community and manuscript could be recommended for publication after minor corrections of comments as follow.
Response:
We are thankful for the encouraging comments/suggestions regarding to our manuscript. We hope that after addressing all these comments and positive criticisms one by one it will improve the understanding and quality of our manuscript for the general readers and scientific community actively involved in intracellular pathogen studies. We incorporated all the changes as suggested and we would surely welcome any further changes that can make our study more readable before final publication.
1. line number 160: Add strain number/catalogue number of BABL/C mice purchase from Charles river lab.
Response: Thanks,
We mentioned the number of BALB/C mice strain and its catalog number of purchase from Charles river lab in line number 164-166 as suggested.
2. Several typos found throughout the manuscript for the use of symbol. Author should correct them. For example: ul need to change to µL (line number 167); symbol for degree (temperature) need to correct.
Response:
Thanks for suggestion. We revised the manuscript several times carefully and corrected all typing mistakes properly. The symbol “ul has been changed into µL” now mentioned in line number 164-165 and all other symbols has been correctly incorporated as suggested. All these corrections are highlighted with red color.
2. line number :160, add sexes of mice used in the studies.
Response:
Thanks for suggestion, in the current study we used female BALB/c mice and we mentioned that in the text in line number 165.
3. How nilotinib were solubilize for experiments? What was the concentration of DMSO treatment in the experiments? Just DMSO label could be read as 100% DMSO which could be toxic.
Response: Thanks for comment.
Initially, we prepared small aliquots of nilotinib by dissolving it in DMSO and stored these aliquots at -80°C to avoid freeze and thaw cycles. At the time of administration each aliquot was obtained from cold storage and diluted into PBS in order to minimized the concentration of DMSO up to 10% containing the final quantity of DMSO was 30µl/animal. Similarly, the uninfected control animals were injected with DMSO diluted in PBS and the final concentration of DMSO was 30µl /animal were injected into each mice as previously reported (Lonskaya et al., 2014) (mentioned in line number 170-172). In addition, for the evaluation of drug toxicity or the toxicity of solvent (DMSO), we performed the histopathological analysis of liver and kidney under microscope after staining with H&E method for all groups as shown in Suppl.Figure 4.
4. References need to format as per guidelines of journal.
Response: Thanks
The references has been formatted as per guidelines of the journal in the revised version of manuscript.
We revised the whole manuscript several times carefully to describe the findings in the text properly and checked spelling, typing and grammatical mistakes. We also added further details in the materials and methods section which might help in understanding the procedure of the experiment. We hope that the revised version of manuscript will be free from any mistakes. The changes we made in the revised manuscript are highlighted with red color. We hope that if any necessary changes needed to be added in the revised manuscript, you will let us know
Round 2
Reviewer 1 Report
The authors have completed a thorough revision, and they have sufficiently addressed all of my comments.